# Rapid diagnostic tests and ELISA for diagnosing chronic Chagas disease: Systematic revision and meta-analysis

**Sandra Helena Suescún-Carrero**[1]*, **Philippe Tadger**[1,2], **Carolina Sandoval Cuellar**[1], **Lluis Armadans-Gil**[3], **Laura Ximena Ramírez López**[1]

**1** Universidad de Boyacá, Tunja, Colombia, **2** Real World Solutions, IQVIA, Zaventem, Belgium, **3** Epidemiology and Preventive Medicine Service, Hospital Universitari Vall d'Hebron—Universitat Autónoma de Barcelona, Barcelona, Spain

* ssuescun27@uniboyaca.edu.co

## Abstract

### Objective

To determine the diagnostic validity of the enzyme-linked immunosorbent assay (ELISA) and Rapid Diagnostic Tests (RDT) among individuals with suspected chronic Chagas Disease (CD).

### Methodology

A search was made for studies with ELISA and RDT assays validity estimates as eligibility criteria, published between 2010 and 2020 on PubMed, Web of Science, Scopus, and LILACS. This way, we extracted the data and assessed the risk of bias and applicability of the studies using the QUADAS-2 tool. The bivariate random effects model was also used to estimate the overall sensitivity and specificity through *forest-plots*, ROC space, and we visually assessed the heterogeneity between studies. Meta-regressions were made using subgroup analysis. We used Deeks' test to assess the risk of publication bias.

### Results

43 studies were included; 27 assessed ELISA tests; 14 assessed RDTs; and 2 assessed ELISA and RDTs, against different reference standards. 51.2 % of them used a non-comparative observational design, and 46.5 % a comparative clinical design ("case-control" type). High risk of bias was detected for patient screening and reference standard. The ELISA tests had a sensitivity of 99% (95% CI: 98–99) and a specificity of 98% (95% CI: 97–99); whereas the Rapid Diagnostic Tests (RDT) had values of 95% (95% CI: 94–97) and 97% (95% CI: 96–98), respectively. Deeks' test showed asymmetry on the ELISA assays.

### Conclusions

ELISA and RDT tests have high validity for diagnosing chronic Chagas disease. The analysis of these two types of evidence in this systematic review and meta-analysis constitutes

**Data Availability Statement:** All relevant data are within the manuscript and its Supporting Information files.

**Funding:** SHSC, LXRL, CSC, PT were funded by the Universidad de Boyacá, Colombia. LAG was funded by the Universitat Autónoma de Barcelona, Barcelona, Spain. The funders had no role in study design, data collection and analysis, decision to publish, or preparation of the manuscript.

**Competing interests:** The authors have declared that no competing interests exist.

an input for their use. The limitations included the difficulty in extracting data due to the lack of information in the articles, and the comparative clinical-type design of some studies.

## Author summary

Chagas disease (CD), an infection caused by the *Trypanosoma cruzi* parasite, affects between 8 and 10 million people worldwide. It is considered one of the main problems of public health in Latin America, and the international migration has caused infected subjects to scatter through the rest of the world, making CD a global health problem. Therefore, it is important to diagnose this infection using laboratory tests, which sometimes becomes a problem due to lack of reference tests and the existence of different types of tests with different sensitivity and specificity values, added to the difficulty in detecting the parasite in its chronic phase. This systematic review and meta-analysis determined the diagnostic validity of the enzyme-linked immunosorbent assay (ELISA) and rapid diagnostic tests (RDT) among individuals with suspected chronic CD. It included 43 studies and concluded that both ELISA and RDTs had adequate diagnostic performance. It is necessary to better understand these two types of diagnostic tests to facilitate clinical decision making about the disease and improve access to treatment for the population at risk.

## Introduction

American Trypanosomiasis or Chagas disease (CD), caused by the protozoan *Trypanosoma cruzi*, continues to be an important cause of illness, disability and death [1]. In recent years, CD has positioned itself as the main parasitic disease in Latin America and as one of the 13 most neglected tropical diseases [2]. It is estimated that about 100 million people are at risk of being infected with *T. cruzi*, in the region, and that there are about 8 to 10 million already infected; with 30,000 new cases per year due to all forms of transmission, which leads to 12,000 annual deaths [3]. In addition, the international migration has caused infected individuals from Latin America to migrate all over the world, which now makes the disease a problem for the global health systems [4].

CD has two forms: acute and chronic. The acute phase is usually asymptomatic or can present as a nonspecific, self-limited febrile syndrome that resolves in approximately 90 % of untreated infected individuals [5]. On the other hand, in its chronic phase, around 60% to 70% of patients do not present any apparent symptoms; 30% of the subjects develop cardiomyopathies with a clinical variety, including arrhythmias, aneurysms, dilated cardiomyopathy, and sudden death [6].

It is essential and important to diagnose *T. cruzi* infection using laboratory tests in order to prescribe the best treatment and, this way, stop the progression of the disease and prevent its transmission [7,8]. However, one limitation is the complexity of the diagnostic process, which is sometimes hampered by the lack of a reference standards, by the availability of multiple types of assays with different sensitivity and specificity values, and by the great difficulty of detecting the parasite in the chronic phase of the disease [9]. The World Health Organization (WHO) recommends using two conventional tests for diagnosing chronic CD, based on different principles and the detection of different antigens. Furthermore, in the case of ambiguous or inconclusive results, a third technique should be used [10]. Thus, serological tests, such as indirect immunofluorescence, indirect hemagglutination, enzyme-linked immunosorbent

assay (ELISA), and immunochromatographic tests or rapid diagnostic tests (RDT) are used [11]. They can be qualitative or semi-quantitative, based on different antigens; some use a multi-epitope antigen and others use a combination of recombinant proteins [12]. The Pan American Health Organization states that the evidence on the validity of tests for diagnosing CD has been considered high in the case of ELISA tests and chemiluminescence analysis, and moderate for RDTs [13]. Each technique has different features in relation to the antigenic targets used, the population evaluated, the cut-off points and the equipment used; therefore, a direct comparison of test performance is more difficult [14]. Taking into account the aforementioned, the purpose of this study was to summarize the evidence available on the diagnostic validity of ELISA and immunochromatographic tests (RDT) in individuals with suspected diagnose of chronic CD.

## Methods

### Protocol and registration

This systematic review and meta-analysis was carried out according to the PRISMA-DTA guidelines (Preferred Reporting Items for Systematic Reviews and Meta-analysis of Diagnostic Test Accuracy Studies -The PRISMA-DTA Statement) [15] for the abstract and the body of the manuscript (S1 and S2 Checklists). The protocol was registered in the PROSPERO database (International Prospective Register of Systematic Reviews) with number CRD42020186588.

### Eligibility criteria

The search included studies that estimated sensitivity and specificity of ELISA or RDT index tests for chronic CD, with participants over five years old, patients with chronic CD, and patients without this disease; studies conducted in endemic and non-endemic areas for CD, that described the reference standards used, studies with a cross-sectional design and a case-control type; written in English, Spanish and Portuguese, published between 2010 and 2020; with research done with volunteers and with samples that included humans. Studies indicating that patients were receiving treatment for CD, those that were related exclusively to acute infection or in newborns, and those with mixed data on patients with acute and chronic infection were excluded.

### Data sources

The databases used for the search, which was carried out from May to August 2020, were: Pubmed/Medline, Scopus; ISIWeb/Web of Science, and LILACS. The corresponding authors of articles included were contacted by email to inquire about missing data or request clarification on studies.

### Study search and selection

The standard search strategy described in *The Joanna Briggs Institute Reviewers' Manual 2015* [16] was used. Thus, there was an initial limited search to identify relevant keywords and indexing terms, followed by a comprehensive search in the databases included with strategies for each of the search engines (S1 Database). Two reviewers (SHSC-LXRL) assessed article titles and abstracts in an independent and blinded manner. Disagreements in the inclusion of studies were resolved by consensus, taking into account that the abstracts should meet the proposed eligibility criteria. Subsequently, the articles were reviewed in full text.

### Data collection process

Two authors (SHSC-LXRL) extracted the following data independently: author(s), year of publication, type of participants, study area, index test, reference test, study period, country of implementation, number of patients and healthy subjects, total number of participants, sensitivity and specificity, risk of bias and applicability.

### Definitions for data extraction

The subjects included in the different studies were classified into: patients who had lived or resided in an endemic area for CD and patients who reside in a non-endemic area.

The study area was considered endemic if CD occurred in this geographic area; and as a non-endemic area, otherwise. The index tests were considered commercial when they were part of a brand of laboratory diagnostic reagents, validated by medical device regulatory agencies and those available on the market; and considered in-house tests when studies indicated that immunoadsorption assays had been designed with different peptides or proteins with the application of non-standard "internal" methods. RDTs are those immunochromatographic assays that throw qualitative results and can be read at first sight.

Reference tests met the standard if they included a combination of serological tests with different antigens detecting antibodies against *T. cruzi*, and an additional test to reach a definitive diagnosis if the results were inconclusive.

The study design was considered clinical-comparative or case-control type if a group of participants diagnosed with chronic CD and a group without this diagnosis had been included; and it was considered non-comparative if a consecutive and representative series of patients with suspected CD had taken the test to be evaluated, as well as the reference test.

### Risk of bias and applicability

Three authors (SHSC-LXLR-CSC) assessed the methodological quality and risk of bias of the studies included, in a blinded and independent manner, using the Quality Assessment of Diagnostic Accuracy Studies-2 (QUADAS-2) tool, which comprises four domains: patient screening, index test, reference test, and flow and time [17]. Each domain was assessed for risk of bias, and the first three domains were also assessed for applicability.

The QUADAS-2 tool was adjusted to the needs of this review, as follows: the risk of bias in patient screening was considered high if a consecutive or random sample of patients had not been used; and unclear if patient recruitment was not specified. The risk of bias related to the index test was considered unclear if there was no specification that the results of the index tests were interpreted without knowing the results of the reference test. The risk of bias related to reference tests was considered high if these tests were interpreted knowing the results of the index test, or if a single reference test had been used (taking into account that the WHO establishes that serological diagnosis in the chronic phase of CD should be based on positive results in two tests that are based on different immunological principles and, in case of inconsistency, on a third test).

### Diagnostic accuracy measures

The reported measures were sensitivity and specificity for each of the index tests assessed for diagnosing chronic CD. When the studies did not have these two measures, they were calculated based on the number of true positives and negatives, as well as on the number of false positives and negatives and the total number of patients.

## Summary of results

Sensitivity and specificity were modeled bivariately with binomial-normal random effects, with a gold standard (GS) assumption, but also with an imperfect gold standard (IGS) model. The GS models were fitted with a Bayesian and classical approach; and the IGS model with a Bayesian approach only. Models were selected with the *deviance information criterion* (DIC) for the Bayesian models, and with the likelihood ratio test for the classical models. Six possible models for the GS were evaluated according to the type of distribution that followed the random effects (normal or mixed normal) and the type of connection (logit, cloglog and probit), and the best model was selected according to the smallest DIC with at least two points difference. The specification of the model with the best fit (in bamdit metadiag) was reproduced in the rest of the packages (meta4diag: Binomial-normal with probit, and metandi and IGS: Binomial-normal with logit) to facilitate comparisons.

The bivariate random effects model was used to estimate the overall sensitivity and specificity and their respective 95 % confidence intervals (CI). The results were plotted in *forest-plots* and ROC space (R DTAplots program), and heterogeneity between studies was assessed visually. R 1.3 *software* (DTAplots, bamdit::plotcompare and meta4diag::meta-regression) [18], Stata 15 (metandi) [19], midas and JAGS were used to conduct the meta-analysis.

## Additional analyses

Meta-regressions were carried out with potential modifiers of diagnostic validity (bamdit plotcompare and meta4diag meta-regression). The variables of interest were study design (clinical comparative or non-comparative), study area (endemic or non-endemic), study risk (low or high risk of bias), sample type (serum, whole blood or not applicable) for the RDTs, and type of test (commercial or in-house) for the ELISA tests but not for the RDTs because of the low number of studies, which made it impossible to estimate them.

All variables were categorized at two levels in both the ELISA and RDT assays to facilitate the comparison of predictive regions and validity estimates. A QUADAS-2 assessment was applied in each study in order to analyze by subgroups. The three levels of the QUADAS-2 became two: low risk and high risk (which included the *high risk and unclear* categories). Of the 7 items of the tool, item 1 (patient screening) and item 3 (reference standard) were considered since they were the only ones with a sufficient number of studies with a high risk of bias. In the rest of items, most studies were low risk.

A sensitivity analysis was carried out excluding influential outliers. Influential studies were reviewed based on the assumption that the subsequent interval distribution of study weight should include one. The publication bias was assessed using Deeks' asymmetry test, which was considered statistically significant with a value of $p < 0.1$ [20].

## Results

### Study selection

As shown in Fig 1, 897 publications were initially identified, of which 739 were eliminated due to duplication in the databases. Of the remaining 158 publications, 75 did not meet the selection criteria in the review by title and abstract. Of the remaining 83 articles, 40 were excluded for the following reasons: 17 due to inadequate study design, 9 did not meet the diagnostic reference test criteria suggested by the WHO, 8 due to non-concordance between the index test and the tests that were to analyze the present investigation (ELISA or RDT), 4 used a population that was in the acute phase of the disease or were studies that analyzed subjects in the

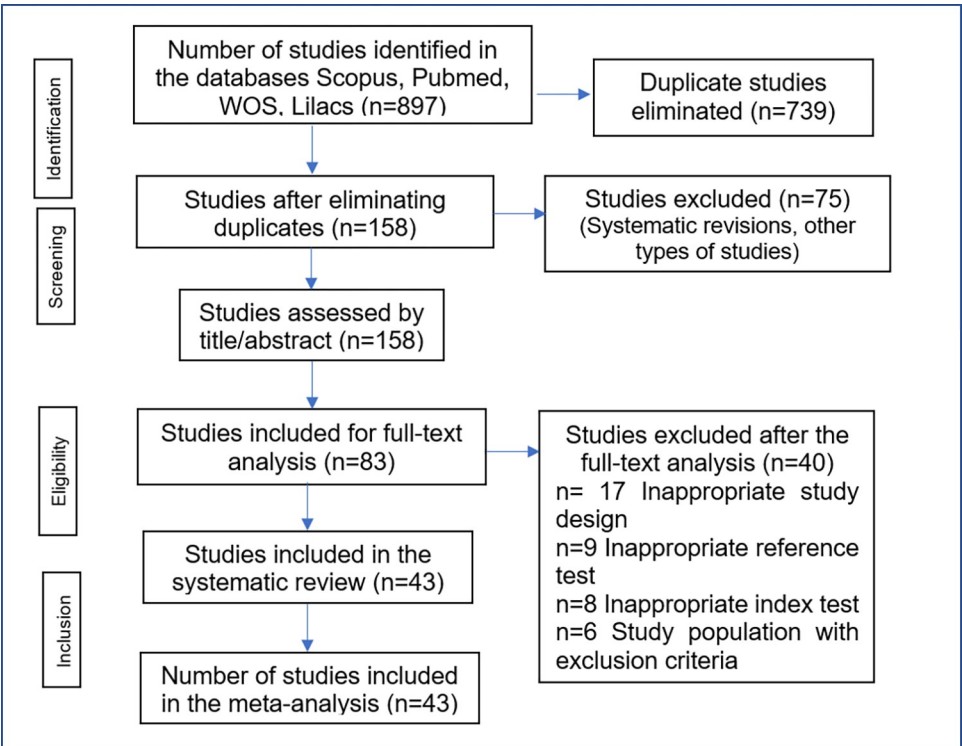

**Fig 1. Study selection.**

acute and chronic phase and 2 included patients with previous treatment for CD. Finally, 43 full-text articles were used for qualitative and quantitative analysis [9,10,21–61].

## Study features

The 43 selected studies were published between 2010 and 2019, with between 3 and 6 studies being published per year, except for 2015, a year in which there were no publications. 27 articles [9,21,22,24,25,27–35,38,41,42,44–46,48,51,53,55,57–59] evaluated ELISA tests; 14 used RDT [10,23,26,32,37,39,40,43,47,50,52,54,60,61] and 2 [36,56] evaluated both tests (Table 1).

In 33 articles (76 for ELISA tests and 39 for RDT), several substudies included aspects such as tests with different peptides, populations from different countries, participation of several reference laboratories or evaluation of different index tests. This data is presented and analyzed by separated in the present investigation. Six articles did not present sensitivity and specificity data [22,24,25,37,38,51] but were included considering that they indicated true and false positive and negative data. A total of 30,356 participants were reported in the 43 articles (from 56 to 10,284 subjects per study).

According to the nature of the sample, 76.7 % of the investigations were carried out with serum [9,21,22,24–30–36,39,41,42,44,45,48–51,53–60], 7% with venous blood [40,46,61], 7% with venous blood and serum [23,43,52], 4.7% with capillary blood [10,37], 2.3% with serum, plasma, venous blood and capillary [47]; and 2.3 % did not report the type of sample used [39].

Of those studies where ELISA tests were assessed, 17 were carried out with *in-house* tests [21,24,26,28,31,35,38,40,44,46,48,52,53,55,60–62]; 12 were commercial trials [9,22,24,25,30,34,36,42,45,48,55,56], and one article studied an *in-house* assay and a commercial assay [26]. The brands assessed that were present in more than one study were Architect

**Table 1. Qualitative analysis of studies selected (ELISA and RDT assays).**

| Ref | Author/year | Type of test | Type of participants | Study area | Index test | Reference test | Study period | Country | Number of sick patients | Number of healthy patients | Total | Sensit. | Specif. |
|---|---|---|---|---|---|---|---|---|---|---|---|---|---|
| 1 (19) | Briceño 2010 | ELISA | Healthy patients and participants with chronic CD and lymphoblastogenesis and Chagasic cardiomyopathy or asymptomatic | Endemic area | AgA-ELISA | Bioschile Ingenieria Genetica S.A Kit, BioKIT and Pharmatest (Laboratorios Pharmatest) Immunofluorescence of epimastigotes Indirect hemagglutination and ELISA) | Not stated | Venezuela | 89 | 477 | 566 | 98.8% | 97%* |
| 2 (30) | Aria 2016_a | ELISA | Blood donors Participants who were positive for syphilis, Hepatitis B and C, HIV, HTLV. | Endemic area | ELISA Chagas test IICS V.1 | Chagatest ELISA recombinant v 3.0 from Wiener. | Not stated | Paraguay | 33 | 23 | 56 | 97% | 91%* |
| | Aria 2016_b | | | | ELISA Chagas test IICS V.1 | BiosChile Test ELISA | | | | | | 97% | 95%* |
| 4 (52) | Llano 2014 | ELISA | Participants with leishmaniasis and heart disease. Symptomatic and asymptomatic | Endemic and Non-endemic area | Chagas (Trypanosoma cruzi) IgG-ELISA (NovaTec Immunodiagnostica GmbH) | ELISA and IFI No brand reported | Not stated | Colombia | 78 | 21 | 99 | 96% | 98%* |
| 5 (56) | Bergmann 2013_a | ELISA | Blood donors from Brazil | Endemic area | Chagatest (Wiener Lab, Argentina). | Immunotransference with TESA antigen (T. cruzi excretor-secretor antigen) No brand reported | Not stated | Brazil | 122 | 39 | 161 | 99% | 99% / 18%* |
| | Bergmann 2013_b | | | | CHAGATEK (Biolab-Mérieux, Rio de Janeiro, Brazil). | | | | | | | 98% | 75%* |
| | Bergmann 2013_c | | | | EIAgen T. cruzi IgG + IgM (Adaltis, Bologna, Italy) | | | | | | | 99% | 53%* |
| 7 (58) | da Silva 2012_a | ELISA | Participants with Chagas disease, indeterminate form, cardiac disorders, digestive disorders with or without cardiac disorders Healthy participants, without Chagas disease, with other diseases such as HIV, hepatitis C, syphilis, visceral leishmaniasis, tegumentary leishmaniasis | Endemic area | ELISA home test IgG whole and subclasses IgG whole: dilution 1:40 | ELISA and indirect hemagglutination assay IHA No brand reported | 2006 | Brazil | 60 | 54 | 114 | 100% | 100% |
| | da Silva 2012_b | | | | ELISA home test IgG IgG 1: dilution 1:10 | | | | | | | 100% | 90.7% |
| | da Silva 2012_c | | | | ELISA home test IgG IgG 2: dilution 1:10 | | | | | | | 100% | 88,9% |
| | da Silva 2012_d | | | | ELISA home test IgG IgG 3: dilution 1:20 | | | | | | | 95% | 98.1% |
| 8 (59) | Pimenta 2019_a | ELISA | Individuals in the rural endemic area | Endemic area | ELISA IBMP-8.1 | ELISA and indirect hemagglutination No brand reported | Not stated | Argentina, Bolivia, Paraguay | 215 | 122 | 337 | 95.3% | 100% |
| | Pimenta 2019_b | | | | ELISA IBMP-8.4 | | | | | | | 100% | 100% |
| 9 (60) | Pierimarchi 2013_a | ELISA | Migrants and blood donors | Endemic and Non-endemic area | ELISA automated using 2 different antigens of T. cruzi. (a recombinant protein and a complete extract of T. cruzi) TcF | In house ELISA, HAI No brand reported BioELISA Chagas, Biokit, Spain (recombinant antigen) | Not stated | Venezuela Italy | 55 | 77 | 132 | 98.1% | 100%* |
| | Pierimarchi 2013_b | | | | ELISA automated using 2 different antigens of T. cruzi. (a recombinant protein and a complete extract of T. cruzi) IMT | Commercial immunomatographic rapid test (Chagas Quick Test, Cypress Diagnostics, Belgium) | | | | | | 96.3% | 100%* |
| 10 (20) | Praast 2011 | ELISA | Blood donors from Germany and patients from Brazil and Guatemala, Bolivia, Argentina and USA | Endemic and Non-endemic area | Abbott Architect Chagas | BioMérieux ELISA cruzi. Wiener Lab test Chagas ELISA recombinant 3.0. Abbott Chagas confirmatory immunoblot assay | Not stated | Germany | 655 | 9629 | 10284 | 99.8% | 99.9% |

(*Continued*)

**Table 1.** (Continued)

| Ref | Author / year | Type of test | Type of participants | Study area | Index test | Reference test | Study period | Country | Number of sick patients | Number of healthy patients | Total | Sensit. | Specif. |
|---|---|---|---|---|---|---|---|---|---|---|---|---|---|
| 12 (8) | Caicedo 2019_a | ELISA | Sera from Instituto Nacional de Salud, public health and department laboratories, blood donors | Endemic area | Test ELISA Chagas III BIOS | ELISA and IFI home test Indirect hemagglutination (Wiener Chagatest HAI) (T. cruzi excretor-secretor antigen) (TESA, bioMérieux Immunoblot) | 2014 to 2016 | Colombia | 256 | 245 | 501 | 99.2% | 97.9% |
| | Caicedo 2019_b | | | | Nonconventional methods Synthetic peptides Umelisa Chagas SUMA | | | | | | | 92.5% | 97.5% |
| | Caicedo 2019_c | | | | Recombinant antigens Architect System Chagas ARCHI | | | | | | | 98.4% | 97.9% |
| | Caicedo 2019_d | | | | BioELISA Chagas BIOKIT | | | | | | | 98% | 94.6% |
| | Caicedo 2019_e | | | | Chagatest ELISA recombinant v. 4 WIENER | | | | | | | 98.8% | 97.9% |
| | Caicedo 2019_f | | | | T. cruzi AB DIAPRO | | | | | | | 95.7% | 97.1% |
| | Caicedo 2019_g | | | | 7.Chagas ELISA IgM + IgG VIRCEL | | | | | | | 99.6% | 97.5% |
| 13 (22) | Caballero 2019_a | ELISA | Blood donors Participants with visceral leishmaniasis, cutaneous and mucocutaneous leishmaniasis, T. rangely, rheumatic fever, toxoplasma, and P. falciparum | Endemic area | ELISA home test with Burunga genotype | TESA-blot Western blot Chagatest Rec v3.0 Wiener kit | Not stated | Panama and Brazil | 40 | 113 | 153 | 100% | 82.3% |
| | Caballero 2019_b | | | | ELISA home test with MM1 genotype | | | | | | | 100% | 77.8% |
| | Caballero 2019_c | | | | ELISA home test with Jose-IMT genotype | | | | | | | 100% | 84.9% |
| | Caballero 2019_d | | | | ELISA home test with Y genotype | | | | | | | 100% | 88.4% |
| | Caballero 2019_e | | | | ELISA home test with FCI genotype | | | | | | | 100% | 81.4% |
| | Caballero 2019_f | | | | ELISA home test with JJ genotype | | | | | | | 100% | 78.7% |
| 15 (24) | Tonelli 2018_a | ELISA | Participants with indeterminate form, chagasic myocarditis, tegumentary leishmaniasis, visceral leishmaniasis | Endemic area | ELISA home test Epitope 1 of B cells derived from the family of Mucin Associated Surface Proteins (MASP) | ELISA (Gold Analisa, Brazil), IHA (Wiener Lab., Argentina) and IFA (Bio-Manguinhos, Brazil) | Not stated | Brazil | 53 | 70 | 123 | 60.3% | 72.8%* |
| | Tonelli 2018_b | | | | ELISA home test Epitope 2 of B cells derived from the family of Mucin Associated Surface Proteins (MASP) | | | | | | | 100% | 97.1%* |
| | Tonelli 2018_c | | | | ELISA home test Epitope 3 of B cells derived from the family of Mucin Associated Surface Proteins (MASP) | | | | | | | 100% | 67.1%* |
| | Tonelli 2018_d | | | | ELISA home test Combination of epitopes 1, 2 and 3 of B cells derived from the family of Mucin Associated Surface Proteins (MASP) | | | | | | | 100% | 82.8%* |
| | Tonelli 2018_e | | | | ELISA home test Combination of epitopes 2 and 3 of B cells derived from the family of Mucin Associated Surface Proteins (MASP) | | | | | | | 100% | 100%* |
| 16 (25) | Pérez 2018 | ELISA | Migrants from Bolivia | Non-endemic area | Architect Chagas | Immunochromatogra-phy tests (ICT) and/or indirect immunofluorescence (IFI) | January 2014-August 2017 | Spain | 307 | 3844 | 4151 | 92.5% | 100% |

(*Continued*)

**Table 1.** (Continued)

| Ref | Author / year | Type of test | Type of participants | Study area | Index test | Reference test | Study period | Country | Number of sick patients | Number of healthy patients | Total | Sensit. | Specif. |
|---|---|---|---|---|---|---|---|---|---|---|---|---|---|
| 17 (26) | Peverengo 2018_a | ELISA | Blood donors | Endemic area | ELISA CP1 [antigens FRA and SAPA (Ags)] | ELISA (Chagatest ELISA) e IHA (Chagatest IHA) from Wiener Lab (Argentina) IFI | Not stated | Argentina | 67 | 67 | 134 | 90.2% | 100% |
| | Peverengo 2018_b | | | | CP3, composed of antigenic determinants MAP, TcD y TSSAII / V / VI | | | | | | | 100% | 92.5% |
| | Peverengo 2018_c | | | | CP1+CP3: | | | | | | | 100% | 100% |
| 20 (29) | Mucci 2017 | ELISA | Healthy patients, patients infected with T.cruzi, in the asymptomatic chronic stage of the disease without heart or gastrointestinal compromise, participants with tegumentary leishmaniasis | Endemic area | ELISA with synthesized peptides | ELISA whole, HAI No brand reported | Not stated | Argentina | 62 | 16 | 78 | 97% | 97% |
| 23 (33) | Neves 2016_a* | ELISA | Serum samples from individuals from CD endemic areas in Pernambuco (Brazil) of the reference laboratory for CD (RLCD, Oswaldo). Cruz Foundation / PE, Brazil) | Endemic area | ELISA of recombinant proteins IBMP8.1 | ELISA recombinant Immuno-ELISA Chagas (Wama Diagnóstica, São Palo, Brazil, batch 14D061), ELISA whole Chagas III (BIOSChile, Ingenieria Genética S.A., Santiago, Chile, batch 1F130525) | Not stated | Brazil | 280 | 20 | 300 | 98.9% | 100% |
| | Neves 2016_b* | | | | IBMP8.2 | | | | | | | 98.2% | 90% |
| | Neves 2016_c* | | | | IBMP8.3 | | | | | | | 95.4% | 95% |
| | Neves 2016_d* | | | | IBMP8.4 | | | | | | | 99.6% | 100% |
| 24 (34) | Neves 2016_a | ELISA | Sera obtained from the serum bank of a reference laboratory of chagasic participants with cardiac disorders. | Endemic area | Immuno-ELISA Chagas (batch 14D061; Wama Diagnóstica, São Paulo, Brazil) and Pathozyme | IFI IFA (Immunocruzi; Biomérieux) Western blot (TESA blot; Biomérieux, Rio de Janeiro, Brazil) | Not stated | Brazil | 186 | 499 | 685 | 97.3% | 100% |
| | Neves 2016_b | | Non-chagasic participants, blood donors with dengue, filariasis, hepatitis B and C, HIV, HTLV, leishmaniasis, leptospirosis, rubella, measles, schistosomiasis and syphilis | | Chagas (batch 7042779; Omega Diagnostics, Scotland, UK), recombinant | | | | | | | 99.5% | 99.2% |
| | Neves 2016_c | | | | Chagas III (batch 1F130525; BIOSChile, Ingenieria Genética S.A., Santiago, Chile) | | | | | | | 100% | 100% |
| | Neves 2016_d | | | | Gold ELISA Chagas (batch CHA132A; Rem, São Paulo | | | | | | | 99.5% | 97% |
| 26 (36) | Reis 2014_a | ELISA | Brazilian chagasic and non-chagasic participants, patients with chagasic heart disease, cutaneous and visceral leishmaniasis | Endemic area | 26_a: ELISA rTc_11623.20 | ELISA recombinant ELISA v. 3.0 kit, Chagatest HAI Wiener Laboratorio Rosario, Argentina IFI Sigma Chemical Company, Missouri, USA Western blot (TESAcruzi, bioMérieux Brazil) | Not stated | Brazil | 58 | 45 | 103 | 94.8% | 98.2% |
| | Reis 2014_b | | | | 26_b: ELISA rTc_N_10421.310 | | | | | | | 89.6% | 94.6% |
| | Reis 2014_c | | | | 26_c: combination of a and b | | | | | | | 95.5% | 98.1% |
| 27 (37) | Izquierdo 2013 | ELISA | Migrants and blood donors Participants with visceral leishmaniasis | Non-endemic area | ELISA chemiluminescent | ELISA whole ELISA ID-PaGIA (DiaMed, Cressier sur Morat, Switzerland) and Chagas Bioelisa Assay (Biokit, Lliçà d'Amunt, Spain) RDT | Not stated | Spain | 92 | 58 | 150 | 100% | 98.3% |
| 28 (38) | Cervantes 2013 | ELISA | Participants with leishmaniasis, tuberculosis, neurocysticercosis, taeniasis and toxoplasmosis | Endemic area | Dot-ELISA | ELISA whole, western blot No brand reported | Not stated | Mexico | 96 | 153 | 360 | 97% | 89% |
| 30 (40) | Iborra 2012 | ELISA | Migrants from endemic countries in Latin America | Non-endemic area | Chemiluminescent immunoassay of microparticles (ARCHITECT Chagas Abbott) | ELISA whole T. cruzi ELISA test system; Ortho Clinical Diagnostic, USA) IFI Immunofluor CHAGAS kit; Biocientífica S.A., Buenos Aires, Argentina Discordance: immunochromatography Onsite Chagas Ab Combo-Cassette (CTK Biotech, Inc. USA). | Not stated | Spain | 76 | 89 | 165 | 100% | 96.6% |

(Continued)

**Table 1.** (Continued)

| Ref | Author / year | Type of test | Type of participants | Study area | Index test | Reference test | Study period | Country | Number of sick patients | Number of healthy patients | Total | Sensit. | Specif. |
|---|---|---|---|---|---|---|---|---|---|---|---|---|---|
| 31 (42) | Longhi 2012 | ELISA | Participants with Kalaazar disease, leishmaniasis, lupus erythematosus, non-chagasic cardiomyopathies, schistosomiasis, juvenile diabetes, idiopathic megaesophagus, and South American blastomycosis | Endemic area | ELISA of T. cruzi and recombinant antigens JL7 | ELISA whole, IFI, HAI No brand reported | Not stated | Brazil | 228 | 108 | 336 | 100% | 95.2% |
| 33 (44) | Hernández 2010_a | ELISA | Patients from La Paz, Bolivia who attended a parasitology laboratory Healthy participants from non-endemic areas such as Germany and Mongolia, and patients with visceral or cutaneous leishmaniasis, syphilis and brucellosis. | Endemic and Non-endemic area | ELISA of fusion polypeptides TcBCDE | ELISA Wiener Chagatest-ELISA Recombinant version 3.0 (Wiener Laboratorios, Santa Fé, Argentina) | Not stated | Bolivia, Brazil, Spain and other European countries | 76 | 54 | 130 | 98% | 97%* |
| | Hernández 2010_b | | Patients from Santa Cruz, Bolivia who attended a hospital | | ELISA of fusion polypeptides TcBCDE | ELISA Wiener Chagatest-ELISA Recombinant version 3.0 (Wiener Laboratorios, Santa Fé, Argentina) Bioelisa Chagas (Biokit, Barcelona, Spain) ELISA recombinant Chagas Stat-Pak (Chembio Diagnostic Systems, Medford, NY), | | | 64 | 21 | 85 | 98% | 94% |
| | Hernández 2010_c | | Patients from Brazil who attended a university hospital | | ELISA of fusion polypeptides TcBCDE | IFI HAI Wiener T. cruzi crude extract ELISA (EIE Biomanguinhos; Fiocruz, Rio de Janeiro, Brazil) | | | 165 | 216 | 381 | 99% | 99% |
| 35 (46) | Dopico 2019_a | ELISA | Serum samples from Latin American pregnant women with toxoplasmosis and Zika. | Non-endemic area | ELISA IBMP 8.1 | ELISA whole ORTHO T. cruzi ELISA Test System (Ortho Clinical Diagnostics Inc., Raritan, USA) | Not stated | Spain | 347 | 331 | 678 | 99.4% | 100% |
| | Dopico 2019_b | | | | ELISA IBMP 8.4 | ELISA recombinant Bioelisa CHAGAS (Biokit S.A., Barcelona, Spain) or BIO-FLASH Chagas (automated chemiluminescent assay; Biokit S.A., Barcelona, Spain) | | | | | | 99.1% | 99.7% |
| 37 (48) | Abras 2016 | ELISA | Participants with leishmaniasis, toxoplasmosis, amoebic liver abscess, malaria, strongyloidiasis, visceral larva migrans, cytomegalovirus, HIV, parvovirus, Epstein Barr, hepatitis B and C, syphilis, and Lyme disease | Non-endemic area | ELISA Chemiluminescent Architect | ELISA whole, recombinant, western blot No brand reported | January 2009 to December 2012 | Spain | 114 | 200 | 314 | 100% | 97.6% |
| 39 (50) | Berrizbeitia 2012_a | ELISA | Participants with leishmaniasis, ascariasis, strongyloidiasis and trichuriasis | Endemic area | ELISA of epimastigotes secretion/excretion proteins Optic density 0,400 | ELISA whole, IFI, HAI No brand reported | Not stated | Venezuela | 50 | 70 | 120 | 100% | 74% |
| | Berrizbeitia 2012_b | | | | ELISA of epimastigotes secretion/excretion proteins Optic density 0,500 | | | | | | | 98% | 80% |
| | Berrizbeitia 2012_c | | | | ELISA of epimastigotes secretion/excretion proteins Optic density 0,600 | | | | | | | 98% | 88% |
| 40 (51) | Neves 2018_a | ELISA | Healthy and chronic CD patients, participants with leishmaniasis | Endemic area | ELISA of chimeric proteins IBMP 8.1 | ELISA whole and recombinant ELISA Chagas III (batch IF130525; BIOSChile, Ingeniería Genética S.A., Santiago, Chile) | Not stated | Brazil, other Latin American countries and the USA | 595 | 526 | 1121 | 96.4% | 99.6% |
| | Neves 2018_b | | | | ELISA of chimeric proteins 8.2 | Immuno-ELISA Chagas (batch 14D061; Wama Diagnostica, São Paulo, Brazil) | | | | | | 93.5% | 99.6% |
| | Neves 2018_c | | | | ELISA of chimeric proteins 8.3 | Pathozyme Chagas (Omega Diagnostics, Scotland, United Kingdom) | | | | | | 96.8% | 100% |
| | Neves 2018_d | | | | ELISA of chimeric proteins 8.4 | Gold ELISA Chagas (Rem, São Paulo, Brazil) | | | | | | 99.4% | 100% |

(Continued)

**Table 1.** (Continued)

| Ref | Author / year | Type of test | Type of participants | Study area | Index test | Reference test | Study period | Country | Number of sick patients | Number of healthy patients | Total | Sensit. | Specif. |
|---|---|---|---|---|---|---|---|---|---|---|---|---|---|
| 41 (53) | Neves 2017_a | ELISA | Participants positive for dengue, filariasis, Hepatitis B and C, HIV, HTLV, leishmaniasis, leptospirosis, measles, rubella, schistosomiasis and syphilis | Endemic area | ELISA of chimeric proteins IBMP 8.1 | ELISA whole Immuno-ELISA Chagas (Wama Diagnostica-SP, Brazil, batch 14D061) | Not stated | Brazil, USA, Mexico, Nicaragua, Guatemala, Honduras and Argentina | 825 | 630 | 1455 | 97.4% | 99.4% |
|  | Neves 2017_b |  |  |  | ELISA of chimeric proteins IBMP 8.2 |  |  |  |  |  |  | 94.3% | 99.6% |
|  | Neves 2017_c |  |  |  | ELISA of chimeric proteins IBMP 8.3 | ELISA Chagas III (BIOSChile, Ingenieria Genetica S.A., Santiago, Chile, batch 1F130525) |  |  |  |  |  | 97.9% | 99.9% |
|  | Neves 2017_d |  |  |  | ELISA of chimeric proteins IBMP 8.4 |  |  |  |  |  |  | 99.3% | 100% |
| 3 (41) | Mendicino 2014_a | RDT | Study conducted in whole bloods to patients that go to primary care clinics in the rural areas of a province of Argentina with clinical suspicion of CD | Endemic area | WL Check Chagas (Wiener Lab SAIC, Argentina) | Chagatest ELISA, Wiener Lab SAIC IHA (IHA Chagas Poly- chaco, Lemos Laboratory SRL, Argentina) IFI in case of discordance. | Not stated | Argentina | 64 | 177 | 241 | 87.3% | 98.8%* |
|  | Mendicino 2014_b |  | Study conducted in serum |  |  |  |  |  | 67 | 171 | 238 | 95.7% | 100%* |
| 6 (57) | Acosta 2013_a | RDT | Participants who were positive for toxoplasmosis, syphilis, tuberculosis, rheumatoid factor and hepatitis | Endemic area | Immunochromatogra-phy test for qualitative detection of IgG anti *Trypanosoma cruzi* | ELISA Chagas test IICS-UNA | Not stated | Paraguay | 97 | 105 | 202 | 97% | 95%* |
|  | Acosta 2013_b |  |  |  | Immunochromatogra-phy test SD Bioline–Korea |  |  |  | 51 | 43 | 94 | 94% | 100%* |
| 11 (21) | Lozano 2019_a | RDT | Inhabitants of the cities Yacuiba and Villa Montes (province of Gran Chaco, department of Tarija) | Endemic area | Chagas Stat-Pak (CSP; Chembio Inc., Medford, USA) | Lysate antigen ELISA from Wiener and recombinant from Wiener Discordance: ELISA (Chagatek, Laboratorio Lemos, Buenos Aires, Argentina) | April to August 2018 | Bolivia | 304 | 381 | 685 | 97.7% | 97.4%* |
|  | Lozano 2019_b |  |  |  | Chagas Detect Plus (CDP; InBIOS International Inc., Seattle, USA) |  |  |  |  |  |  | 98.4% | 87.1%* |
| 14 (23) | Mendicino 2018_a | RDT | Inhabitants of the northern province of Santa Fe, located in Gran Chaco | Endemic area | RDT A: WL Check Chagas (Wiener Lab SAIC, Argentina) | IHA (IHA Chagas Polychaco, Lemos Laboratory SRL) ELISA (Chagatest ELISA, Wiener Lab SAIC) When results were discordant, IFI was performed with commercial conjugates and smear prepared with epimastigotes of *T. cruzi* of the strain Tulahuen 0.10 | Not stated | Argentina | 42 | 64 | 106 | 90.5% | 100%* |
|  | Mendicino 2018_b |  |  |  | RDT B: SD BiolineChagasAb Rapid (Standard Diagnostics Inc., Korea) |  |  |  |  |  |  | 97.6% | 93.8%* |
| 19 (28) | Navarro 2011 | RDT | Migrants in Spain from Bolivia, Ecuador, Peru | Non-endemic area | Rapid immunochromato-graphy test (ICT) (Simple Chagas WB, Operon) | IFI, ELISA | May 2008 to December 2009 | Spain | 57 | 219 | 276 | 88% | 94%* |
| 21 (31) | Angheben 2017 | RDT | Migrants and travelers from endemic countries of Latin America | Non-endemic area | Rapid immunochromato-graphy test Chagas Quick Test (CQT), Cypress diagnostics of Belgium | ELISA whole ELISA for Chagas III®, BioChile, Chile (Lys ELISA) ELISA recombinant Bio-Elisa Chagas, Biokit, Spain (Ric-ELISA). | April 2009 to June 2015 | Italy | 256 | 384 | 640 | 82.8% | 98.7% |
| 22 (32) | Egüez 2017_a | RDT | People who went to the reference laboratory Laboratorio de Referencia Departamental Chuquisaca in Sucre | Endemic area | Chagas Stat-Pak (CST; Chembio Inc., Medford, USA) | Elisa whole Wiener Lab (Rosario, Argentina), ELISA Wiener v2.0 | March to May 2014 | Bolivia | 209 | 133 | 342 | 87% | 93.2% |
|  | Egüez 2017_b |  |  |  | Chagas Detect Plus (CDP; InBios Inc., Seattle, USA) | ELISA recombinant ELISA Wiener v3.0 HAI IHA test, Chagas Polychaco kit from laboratorio Lemos (Buenos Aires, Argentina) |  |  |  |  |  | 93.4% | 95.2% |
| 25 (35) | Shah 2014_a | RDT | Participants with early and advanced heart disease | Endemic area | Chagas Detect Plus (CDP) (InBios International Inc, Seattle) Serum | ELISA recombinant Wiener recombinant v3.0 ELISA IFI does not report any brand HAI IHA (Chagas Polychaco kit; Lemos Laboratories, Buenos Aires, Argentina | April-May 2013 | Bolivia | 292 | 293 | 585 | 96.2% | 98.8% |
|  | Shah 2014_b |  |  |  | Chagas Detect Plus (CDP) (InBios International Inc, Seattle) Blood |  |  |  |  |  |  | 99.3% | 96.9% |

(*Continued*)

**Table 1.** (Continued)

| Ref | Author / year | Type of test | Type of participants | Study area | Index test | Reference test | Study period | Country | Number of sick patients | Number of healthy patients | Total | Sensit. | Specif. |
|---|---|---|---|---|---|---|---|---|---|---|---|---|---|
| 29 (39) | Flores 2012_a | RDT | Migrants and travelers to endemic countries of Latin America, natives and people born in Spain, with epidemiologic risk factors. Participants with visceral leishmaniasis and malaria. | Non-endemic area | Operon Immunochromatography test (ICT-Operon; Simple Stick Chagas serum and plasma | ELISA whole, IFI, PCR Does not report any brand | Not stated | Spain | 63 | 188 | 251 | 100% | 92.6% |
| | Flores 2012_b | | | | Simple ChagasWB Operon S.A., Spain) Peripheral blood | | | | | | | 91.8% | 93.7% |
| | Flores 2012_c | | | | Simple ChagasWB Operon S.A., Spain) Capillary blood | | | | | | | 86.1% | 94.8% |
| 32 (43) | Barfied 2011_a | RDT | No patient characteristics reported | Endemic area | Chagas STATPAK from Laboratorio Lemos, Argentina 15 minutes | ELISA whole BioMerieux ChagaTek ELISA ELISA recombinant Laboratorio Lemos Biozíma Chagas recombinant | Not stated | Argentina | 190 | 185 | 375 | 95.3% | 99.5% |
| | Barfied 2011_b | | | | Chagas STATPAK from Laboratorio Lemos, Argentina 20 minutes | | | | | | | 95.8% | 99.5% |
| | Barfied 2011_c | | | | PATH Rapid Test from Lemos 15 minutes | | | | | | | 97.9% | 96.2% |
| | Barfied 2011_d | | | | PATH Rapid Test from Lemos 20 minutes | | | | | | | 99.5% | 96.8% |
| | Barfied 2011_e | | | | PATH Rapid Test from Lemos 25 minutes | | | | | | | 98.9% | 94% |
| 34 (45) | Chapouis 2010_a | RDT | Latin American migrants from several countries where the disease is endemic. | Non-endemic area | Stat-Pak in blood | ELISA whole and recombinant Stat-Pak assay: the bioMerieux Elisa cruzi Biokit bioelisa Chagas | June to November 2008 | Switzerland | 125 | 874 | 999 | 95.2% | 99.9% |
| | Chapouis 2010_b | | | | Stat-Pak in serum | | | | | | | 96% | 99.8% |
| 36 (47) | Reithinger 2010_a | RDT | Patients from Argentina | Endemic area | Trypanosoma Detect MRA rapid test; Inbios, Seattle, WA in | ELISA whole, IFI, HAI Does not report any brand | 2000 to 2007 | United Kingdom | 40 | 61 | 101 | 82.5% | 96.7% |
| | Reithinger 2010_b | | Patients from Ecuador | | Trypanosoma Detect MRA rapid test; Inbios, Seattle, WA in | Chagas III (BiosChile Ingeniería Genética S.A., Santiago, Chile) Chagatek ELISA (bioMérieux, Buenos Aires, Argentina) IFI, HAI does not report any brand | | | 51 | 49 | 100 | 84.3% | 95.9% |
| | Reithinger 2010_c | | Patients from México | | Trypanosoma Detect MRA rapid test; Inbios, Seattle, WA in | Kit Chagas III (BiosChile Ingeniería Genéíca S.A., Santiago, Chile) IFI, HAI does not report any brand | | | 40 | 60 | 100 | 77.5% | 100% |
| | Reithinger 2010_d | | Patients from Venezuela | | Trypanosoma Detect MRA rapid test; Inbios, Seattle, WA in | ELISA home test IFI, HAI does not report any brand | | | 40 | 25 | 65 | 95% | 100% |
| 42 (54) | Sánchez 2014_a | RDT | Existing samples from the serum banks in each national reference laboratory | Endemic area | Rapid test from different commercial brands: OnSite Chagas Ab Rapid test | ELISA whole, IFI, HAI Does not report any brand | Not stated | Argentina, Brazil, Colombia, Costa Rica and Mexico | 237 | 237 | 474 | 90.1% | 91% |
| | Sánchez 2014_b | | | | WL Check Chagas | | | | | | | 88.7% | 97% |
| | Sánchez 2014_c | | | | Trypanosoma Detect Rapid Test | | | | | | | 92.9% | 94% |
| | Sánchez 2014_d | | | | Chagas Quick Test | | | | | | | 92.9% | 93.2% |
| | Sánchez 2014_e | | | | Chagas Stat-Pak assay | | | | | | | 87.2% | 93.2% |
| | Sánchez 2014_f | | | | SD Chagas Ab Rapid | | | | | | | 90.7% | 94% |
| | Sánchez 2014_g | | | | Serodia Chagas | | | | | | | 94.2% | 94.7% |
| | Sánchez 2014_h | | | | ImmunoComb II Chagas Ab | | | | | | | 97.2% | 94% |

*(Continued)*

**Table 1.** (Continued)

| Ref | Author / year | Type of test | Type of participants | Study area | Index test | Reference test | Study period | Country | Number of sick patients | Number of healthy patients | Total | Sensit. | Specif. |
|---|---|---|---|---|---|---|---|---|---|---|---|---|---|
| 43 (55) | López 2010 | RDT | Adult patients from Central and South America who went to the Primary Care Center Clot in Barcelona for a CD screening. | Non-endemic area | Simple CHAGASWB (Operon S. A, Spain) | ELISA whole (ELISAc) home test ELISA recombinant Bioelisa Chagas, BiokitS.A., Spain Western blot | Not stated | Spain | 49 | 92 | 148 | 92.5% | 96.8% |
| 18 (27) | Antinori 2018_a | ELISA and RDT | Migrants from Latin America (Brazil, Bolivia, Ecuador; Salvador and Peru) who participated in health promotion programs and screening | Non-endemic area | (ARCHITECT Chagas, Abbott, Chicago, IL, USA.) | Unknown reference standard | July 30, 2013 to July 30, 2014 | Italy | 48 | 453 | 501 | 100% | 99.7% |
| | Antinori 2018_b | | | | (BioELISA Chagas III, BiosChile, Santiago, Chile) | | | | | | | 95.9% | 99.7% |
| | Antinori 2018_c | | | | (Trypanosoma cruzi IgG Rapid Test, ImmunoSpark, SD, Rome, Italy) | | | | | | | 89.2% | 92.5% |
| 38 (49) | Whittman 2019_a | ELISA and RDT | Participants who are blood donors at the American Red Cross- migrants and native American | Non-endemic area | Ortho ELISA | ELISA whole ELISA Ortho ELISA ELISA recombinant Abbott PRISM (Abbott Laboratories, Abbott Park, IL) Radioimmunoprecipi-tation RIPA, Quest Diagnostics (Chantilly, VA) Immunochromatogra-phy Abbott enzyme strip assay (ESA) | September 2006 to June 2018 | USA | 500 | 300 | 800 | 92.4% | 100% |
| | Whittman 2019_b | | | | Hemagen ELISA | | | | | | | 88% | 100% |
| | Whittman 2019_c | | | | Wiener ELISA | | | | | | | 94% | 99.3% |
| | Whittman 2019_d | | | | InBios rapid test | | | | | | | 97.4% | 92.3% |

* The authors of this paper calculated the sensitivity and specificity

Sensit: Sensitivity

Specif: Specificity

Chagas Abbott [9,30,34,36,48,55], ELISA Chagas III BIOSChile [9,36,41], and Chagatest Wiener [9,25]. 11 different RDTs were assessed; the most used commercial brands were Chagas Detect Plus InBios [10,40,43,54,56], Chagas Stat-Pak assay ChemBio [10,40,50,52,60], and WL Check Chagas Wiener [23,32,60]. Of the 16 studies that analyzed RDTs, 8 did not report who read the results [23,26,36,37,40,54,60,61], 4 did not specify whether or not they had training in reading this type of test [10,47,50 and 56] and 4 studies indicated that the reading was done by qualified personnel [32,39,43,52].

62.7 % of the studies were carried out in countries with endemic regions and native population such as Brazil, Argentina, Bolivia, Venezuela, Paraguay, Mexico, Colombia, Guatemala, Panama, Ecuador, Peru, Nicaragua, Honduras and Costa Rica. 27.9 % of the studies were carried out in non-endemic regions, with a migrant population from countries such as Spain, the United States, Italy, Switzerland and the United Kingdom.

51.2 % of the studies had a non-comparative observational design (cohort or cross-sectional) [10,23,27,32,36–41,43,44,46,48–53,57–59]; 46.5 % were comparative clinical design studies (case-control) [9,21,22,24–26,28–31,33–35,42,45,47,54–56,60] and 2.3 % corresponded to mixed studies (comparative and non-comparative) [61].

Of the 43 studies analyzed, 18 of them [21,22,27,31,33,38,42,44–47,49,51,53,55,57–59] included individuals with other diseases to assess cross-reaction. These diseases were infectious parasitic diseases such as cutaneous, mucocutaneous, visceral and tegumentary leishmaniasis, malaria, toxoplasmosis, schistosomiasis, strongyloidiasis, filariasis, neurocysticercosis, taeniasis, ascariasis, trichuriasis, amoebic liver abscess, visceral larva migrans, and patients positive for *Trypanosoma rangeli*; bacterial, such as syphilis, rheumatic fever, leptospirosis, tuberculosis, brucellosis, and Lyme disease; viral infections such as hepatitis B and C, HIV, rubella, measles, dengue, Zika, parvovirus, HTLV, Epstein-Barr, or cytomegalovirus; fungal infections, such as South American blastomycosis; and other non-infectious diseases, such as lymphoblastogenesis, juvenile diabetes, lupus erythematosus, and idiopathic megaesophagus. However, only 13 of these [21,26,27,29,31,38,42,45–47,55,58,59] reported cross-reaction percentages, which ranged between 0% and 62.5%, and whose main reaction was related to *Leishmania spp*.

## Risk of bias and applicability

The quality assessment of the studies included in the analysis of ELISA tests is shown in S1 and S2 Figs; and those of RDTs are shown in S3 and S4 Figs. The risk of bias was assessed in the four domains:

1. Patient selection was assessed at high risk of bias in 19 articles for ELISA tests and in 6 for RDT because a consecutive or random sample of patients was not used, since it was unclear in 5 articles for ELISA tests, and because it did not specify patient recruitment; and it was assessed as low risk in 9 studies for RDTs.

2. The risk of bias related to the index test was assessed as unclear in 19 studies for ELISA and in 7 for RDTs because they did not state clearly whether the index test results were interpreted without knowledge of the reference standard results.

3. The bias related to the reference test had 19 studies for ELISA tests and 7 for RDTs, which were classified as high risk because the result of the reference test was interpreted knowing the results of the index test, or a single diagnostic test was used as a reference standard ([62] [63],).

4. Flow and time were assessed as high risk of bias in four studies for ELISA tests, as not all patients received the same reference standard; while all included studies for RDTs were found to be at low risk of bias on this dimension.

Regarding the applicability in the first three domains, 100% of the articles that assessed both ELISA and RDTs tests were classified as low risk because they coincided with the review question.

## Synthesis of results

**Selecting the model for Elisa and Rapid Diagnostic Tests.** The Bayesian model (GS) that best fit the analysis of ELISA tests was the binomial-normal with the *logit* or the *probit* as the link function (DIC = 630 and DIC = 631, which can be considered equivalent [S1 Table]). Similarly, for the RDT analysis, with the binomial-normal *probit* model (DIC = 416 [S2 Table]); thus, a *probit* link was used for the two tests analyzed.

**Sensitivity and specificity of ELISA tests.** The ELISA tests had an overall sensitivity of 99% (95% CI: 98–99) and an overall specificity of 98% (95% CI: 97–99) (Fig 2). Some studies presented outliers in sensitivity [33] and specificity [25]. In the predictive region, greater variability is observed for specificity than for sensitivity (Fig 3), therefore, it is observed that there is greater heterogeneity in specificity than in sensitivity.

**Sensitivity and specificity of Rapid Diagnostic Tests.** The RDTs had an overall sensitivity of 95% (95% CI: 94–97), and an overall specificity of 97% (95% CI: 96–98). There were studies with atypical values or outliers [10,54] (Those that did not follow the patron of most of

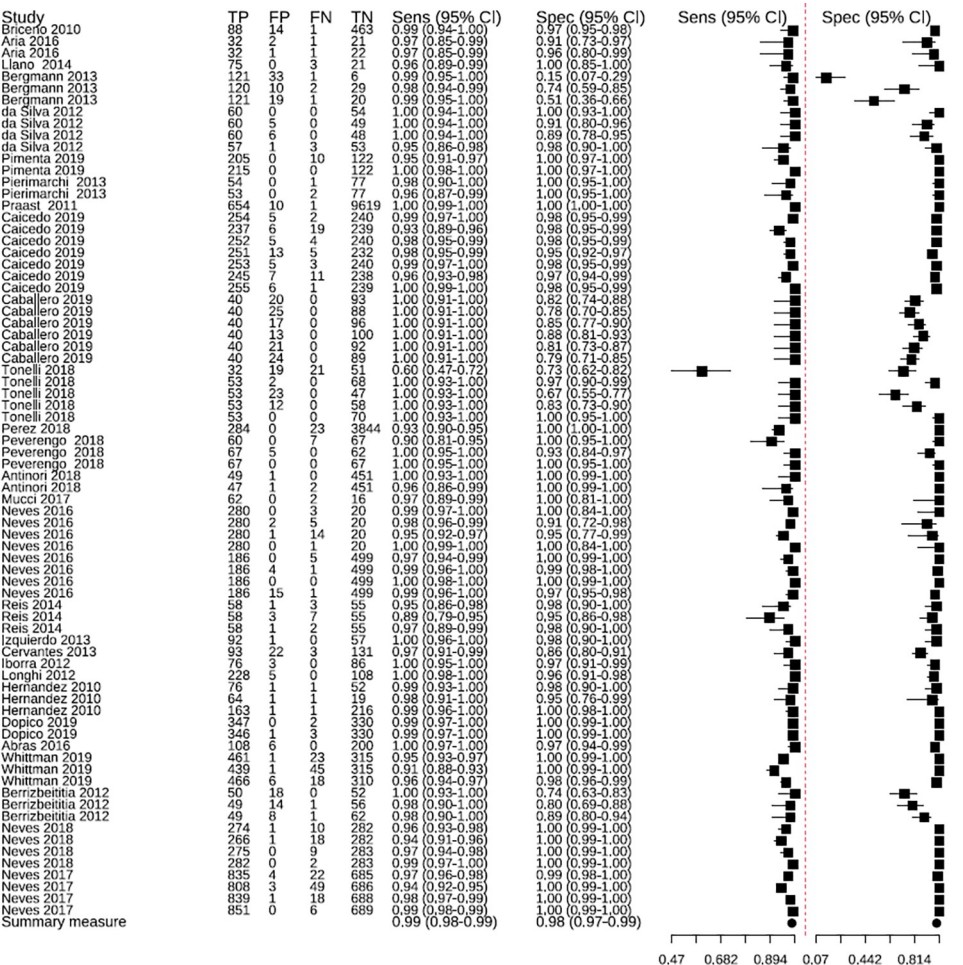

**Fig 2. Sensitivity and specificity of ELISA tests in the studies included in the meta-analysis.** TP: true positives; FP: false positives; FN: false negatives; TN: true negatives.

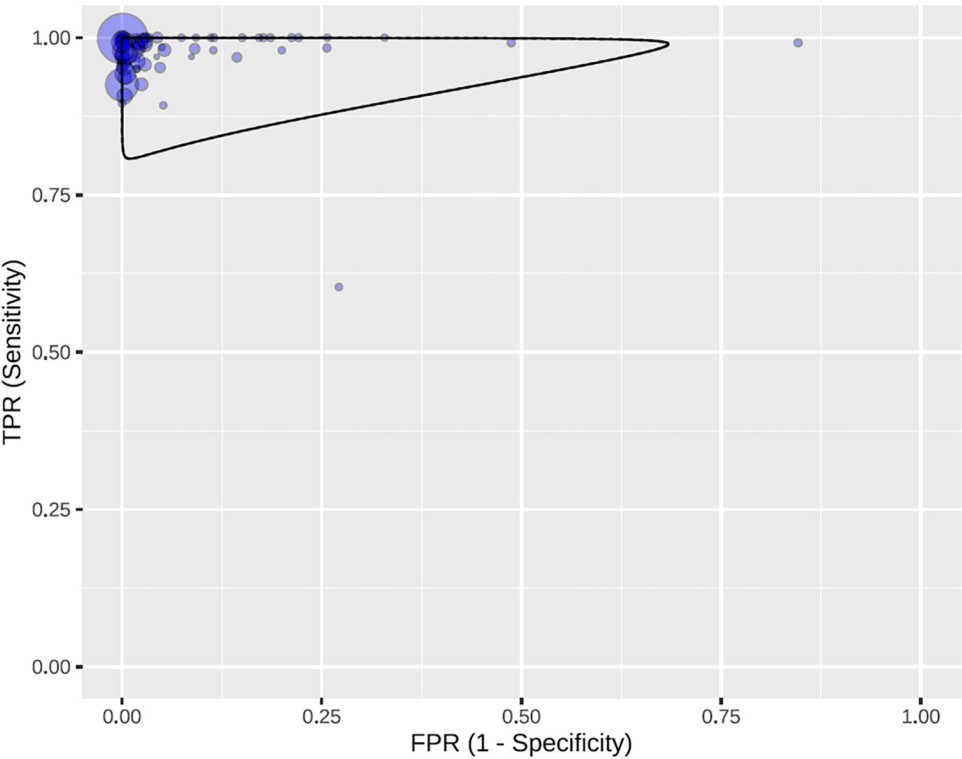

**Fig 3. Predictive region in the ROC space, with all the studies for ELISA tests.**

the studies, which means that they were strayed of the tendency) (Fig 4). The predictive region (Fig 5) is more symmetric, and a slightly higher variability is observed for sensitivity. Therefore, it is identified that there is a slightly greater heterogeneity for sensitivity.

**Subgroup analysis.**   For the ELISA tests, sensitivity estimates were similar by subgroups. The punctual estimate ranged from 98% to 99%, and CIs ranged from 97% to 99%. As for specificity, the sensitivity value ranged between 95% and 100%; the subgroup with low risk of bias was the one that showed the lowest specificity, with 95% (95% CI: 91–97), and the non-endemic area was the one that reported the highest specificity, 100% (95% CI: 99–100) (S5 Fig).

Regarding the ELISA tests, the moderators that showed the greatest difference in the predictive region were study design (clinical comparative or non-comparative), where the comparative studies presented low heterogeneity, while the non-comparative studies had greater heterogeneity; and the study risk subgroup (low or high risk of bias), because the studies are similar when they present risk, showing a lower precision (S6 Fig).

For the RDTs, the sensitivity estimates by subgroups vary more than the specificity. The punctual sensitivity estimates ranged from 91% to 97%, and CIs ranged from 86% to 98%, with the most notable difference (with no overlapping CIs) also observed between the non-comparative design subgroup, with 97 % (95 % CI: 96–98), and the comparative clinical design, with 93 % (95 % CI: 90–95). The punctual specificity estimate ranges from 95% to 98%, and CIs range from 93% to 99%. The most notable difference (without overlapping CIs) is between the non-comparative design subgroup, with 98% (95% CI: 97–99), and the clinical comparative design, with 95% (95% CI: 93–99) (S7 Fig).

Regarding the RDTs, the moderators that show the greatest difference in the predictive region are the subgroups study area (endemic or non-endemic), sample type (serum, whole blood or not applicable), and study risk (low or high risk of bias) (S8 Fig).

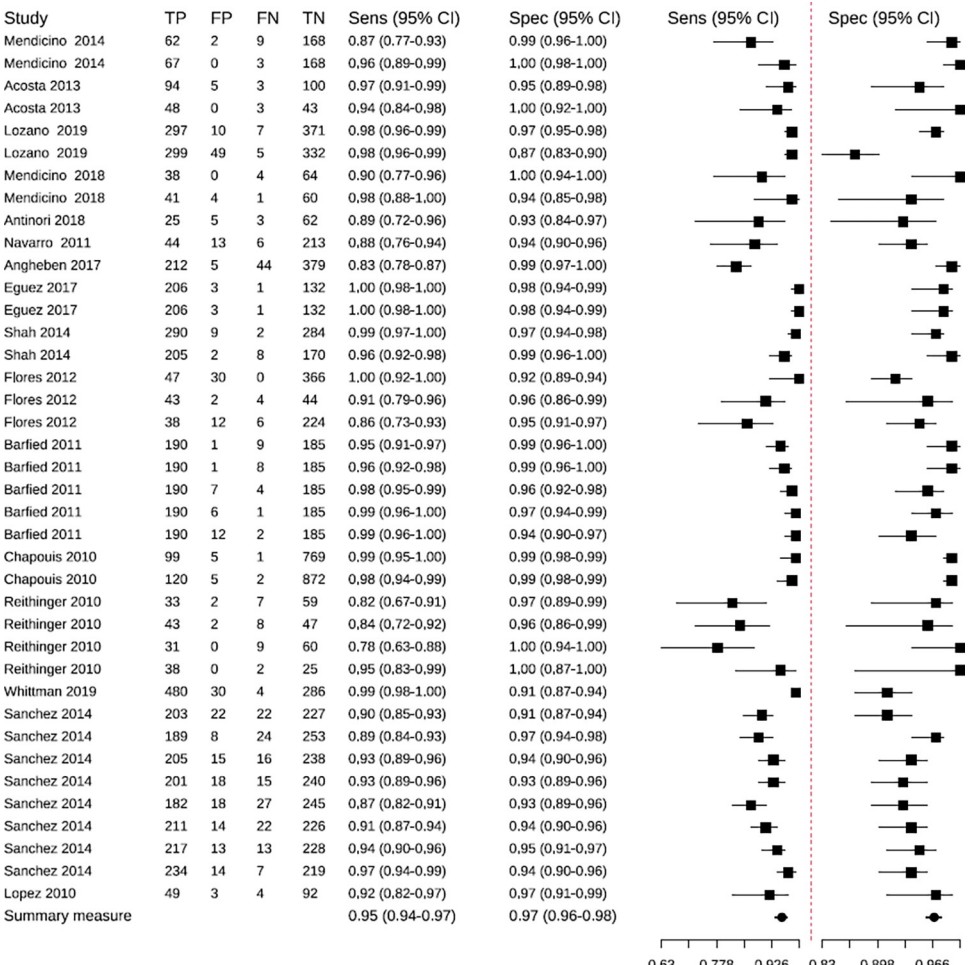

**Fig 4. Sensitivity and specificity of Rapid Diagnostic Tests in the studies included in the meta-analysis.** TP: true positives; FP: false positives; FN: false negatives; TN: true negatives.

**Analysis of influential observations.** Four [9,25,33,34] and five [10,23,40,52,54] were the most influential for the ELISA test studies and the RDT respectively. For each test, two models were fitted (one with all studies and one for all studies except the influential ones) in order to observe the effect of excluding influential studies on the accuracy and predictive region. Both models showed a similar predictive region. The influential studies of ELISA tests were always studies with a clinical-comparative design; whereas the ones for RDTs were non-comparative studies (S9 Fig).

**Publication bias.** Asymmetry was observed in the *funnel plots* for ELISA tests; whereas no asymmetry was observed in the RDTs. The result of the Deeks' asymmetry test was statistically significant for ELISA tests (p < 0.01), but not for RDTs (p = 0.64).

## Discussion

### Summary of evidence

The combined sensitivity and specificity of ELISA tests were 99% (95% CI: 98–99) and 98% (95% CI: 97–99); whereas the ones for RDTs were 95% (95% CI: 94–97) and 97% (95% CI: 96–98). The overall sensitivity of RDTs was lower than that of ELISA tests. According to the results

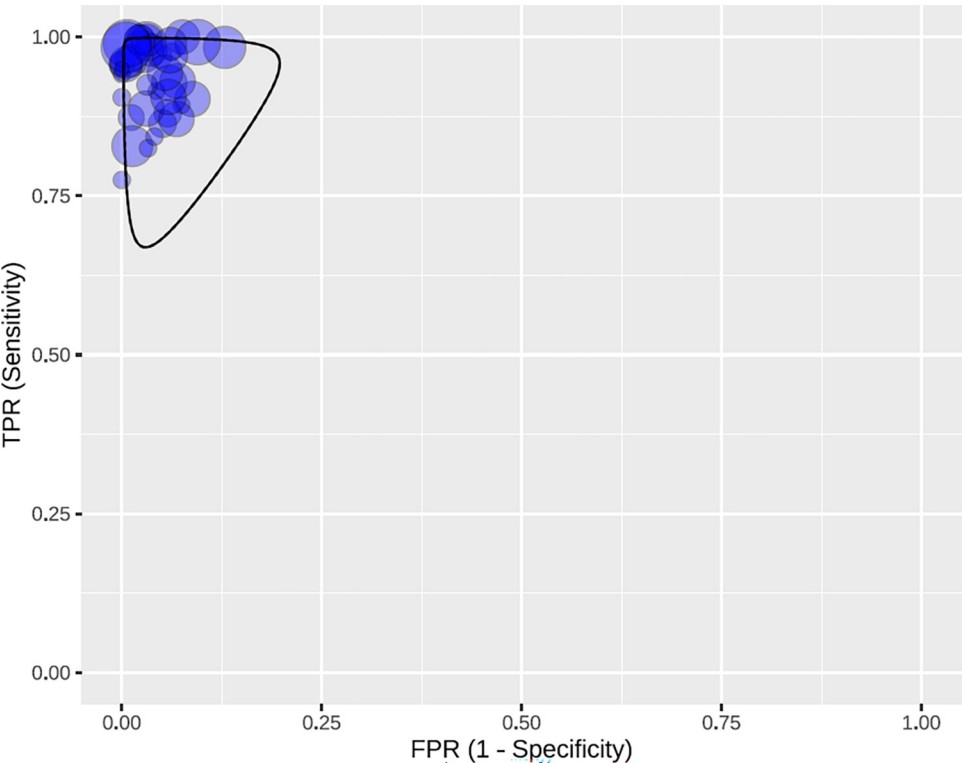

**Fig 5. Predictive region in the ROC space, with all the studies for RDTs.**

obtained in this meta-analysis, the sensitivity and specificity of ELISA tests were higher than those reported in another meta-analysis, in which ELISA tests were compared with RDTs [64] and in which the sensitivity was 97.7. % (96.7%-98.5%) and specificity 96.3% (94.6%-97.6%). The sensitivity of ELISA tests was also higher than in another meta-analysis, where different types of tests were analyzed for diagnosing CD and in which they obtained a sensitivity result of 90% (89%-91%) and a specificity of 98%. % (98 %-98 %) [65], as in this study. Regarding RDTs, the sensitivity and specificity data obtained in this study were lower than those documented in another meta-analysis, where they included clinical trials with recruitment of cohorts of individuals at risk of exposure to *T. cruzi*, which were 96.6 % (CI95%: 91.3–98.7) and 99.3 % (CI95 %: 98.4–99.7), respectively [66].

Regarding heterogeneity, in the ELISA tests it was identified that this measure was higher in specificity than in sensitivity, which is similar to that reported by Brazil [64], while in the RTDs, the heterogeneity was limited, being slightly higher for sensitivity, coinciding with what was described in the systematic review carried out by Angheben [39], results that differ from those reported by Afonso [65].

This meta-analysis showed greater specificity for ELISA tests in non-endemic areas, and greater sensitivity for endemic areas. The diagnostic performance of RDTs was the same for endemic and non-endemic areas, unlike what Angheben et al. [66] reported, where performance was higher in endemic than non-endemic areas.

No differences in sensitivity were observed between the *in-house* and commercial ELISA tests. The same showed to be true for specificity in the two types of tests. This differs from what was stated in the meta-analysis by Afonso et al. [65], where the commercial tests were more sensitive than the *in-house* ones; however, the specificity was similar between the two, as it was observed in this meta-analysis.

All ELISA test assays were made on serum, with the exception of the study by Cervantes-Landín et al. [46], who used venous blood impregnated on filter paper before its coagulation. However, RDTs were made on all types of samples, mostly serum [23,26,32,36,50,54,56,60], but also venous blood and serum [23,43,52]; venous blood [40,61]; capillary blood [10,37]; serum, venous blood, and capillary blood [47], and one study did not specify the sample type [39]. These RDTs are frequently used in *Point-of-Care* tests (immediate diagnostic analyses), which are performed outside the laboratory, closer to the patient, with easily transportable material and equipment, and which results are available in minutes or in less than one hour. Its application is greater in developing countries [67] since the use of samples such as capillary blood -easy to collect without requiring collection tubes or centrifugation-, would be the best choice for RDTs. The results obtained in this study, depending on the type of sample, allow us to infer that the diagnostic performance of the RDTs was good, regardless of the type of sample.

The trypanosomatids that affect men in America belong to the *Leishmania and Trypanosoma* genera. ELISA tests have been valuable for diagnosing these two agents, but their specificity may be low due to the cross-reactivity between the two species of parasites; thus, it is important to take this into account, especially when you want to know the prevalence of these two diseases in endemic areas [68]. Furthermore, most of the areas where *T. cruzi* is found are co-endemic for Le*ishmania sp*. and *T. rangeli*, which complicates the diagnosis of CD [69]. This coincides with what was documented in 13 of the studies analyzed, according to which the percentage of cross-reaction is between 0% and 62.5% with various diseases, among which leishmaniasis stands out.

## Limitations

The items in which a risk of bias was detected in more articles were patient screening and reference standard for the two types of tests assessed; furthermore, the risk of bias associated with the index test was unclear in more than 50% of the studies for ELISA tests. On the other hand, in the applicability items, the studies selected did not present a risk, similar to that described in the meta-analysis carried out by Angheben et al. [66], in relation to the quality of the RDT tests. For the ELISA tests, the sensitivity in the studies with low or high risk of bias was very similar, different from that described by Afonso et al. [65], who reported higher sensitivity in low-risk studies than in high-risk ones. As for specificity, it was higher in the articles that presented a high risk of bias, different from what was reported in this same meta-analysis, where the studies with low risk showed greater specificity.

46.5% of the studies were found to be comparative, and 2.3% mixed studies (comparative and non-comparative). This influences the quality of the studies included because the best design to assess the validity of diagnostic tests is the non-comparative one (cohort or cross-sectional), where a consecutive and representative series of patients with suspected disease are given the test to be evaluated and the reference test in a blind and independent manner and interpreted in the absence of any additional clinical information, which will not be available when the test is used in the practice, either.

In the literature found on this topic, most studies evaluating diagnostic tests take into account the blinding of the reference test results, but few of them have used this non-comparative design [70]. In this meta-analysis, lower sensitivity and specificity values have been found in RDT studies with a comparative clinical design, where a group of patients diagnosed with the disease and a group without this diagnosis were included. This is an unexpected result since studies with a clinical comparative or case-control design tend to overestimate sensitivity and specificity; however, it is possible that this effect only occurs when severe cases are

included in the case group [71]. Additionally, a high risk of publication bias was identified for ELISA tests, which coincides with the study by Afonso et al. [65].

The meta-analysis could have been affected by the search made in only four databases because not all studies related to the topic might have been included. Regarding the studies included, ELISA and RDT tests were not compared because only one study [36] had done so; the same was true for the *in-house* and commercial RDT subgroups which could not be assessed since only one study [26] included *in-house* RDTs.

The quality of the systematic review was also influenced by the results of the selected articles and their design (some of which were clinical-comparative); that is, some variables could not be explored in the subgroup analysis because they were not reported in many studies, for instance, study period, cross reactions, distribution by rural and urban area, generation or version of diagnostic tests, estimation of cut-off points, geographic areas of the strains used as sources of antigens, and the type of antigen (multiepitope or combination of recombinant proteins).

Subsequent studies should follow the instructions given by the WHO and carry out two tests in parallel using different antigens due to the immunogenic diversity of the different strains of the parasite, the immune response of the patients, and the existence of cross-reactions with other trypanosomatids that coexist in endemic areas [72]. Likewise, a sufficient number of samples should be included to evaluate the cross-reactions between chronic CD and Leishmania infection, considering that patients with either of the two infections, or with mixed infection, may be misdiagnosed given the crossed serological reactions when combinations of uncharacterized antigens are used [22].

Regarding the current status of the implementation of RDTs for diagnosing chronic CD in endemic areas of Latin America, these are used as tests of choice in screening programs for its detection, early treatment and control, and they represent a first approach at point of care for the rapid diagnosis of CD in endemic countries [73]. In addition, since RDTs are easier to use than ELISA, it would be feasible to use them more often in screening programs, which would facilitate the detection of CD cases without ignoring the current recommendations to confirm positive results through conventional methods.

## Conclusions

According to this systematic review, ELISA and Rapid Detection Tests (RDTs) have a high validity for diagnosing chronic CD; however, the overall sensitivity of the second test was lower than that of the first one, so it is important to better study the variables that influence the validity of the RDTs, some of which had not been taken into account in some of the studies included.

The usefulness of RDTs for screening CD in epidemiological contexts, such as endemic regions that are difficult to access or non-endemic regions with a high prevalence of chronic CD, should also be assessed, as well as the inclusion of RDTs in the diagnostic algorithms used for its detection, in order to improve access to treatment since the first level of primary health care.

## Supporting information

**S1 Checklist. PRISMA-DTA for abstracts checklist.**
(DOCX)

**S2 Checklist. PRISMA-DTA checklist.** Checklist for reporting of systematic reviews and meta-analysis of diagnostic test accuracy studies.
(DOCX)

**S1 Database. Database search strategy.**
(DOCX)

**S1 Table. ELISA test. Bivariate model goodness.**
(DOCX)

**S2 Table. RDTs Bivariate model goodness.**
(DOCX)

**S1 Fig. Risk of bias and applicability of ELISA tests.**
(TIFF)

**S2 Fig. Summary of risk of bias and applicability of ELISA tests.**
(TIFF)

**S3 Fig. Risk of bias and applicability of RDTs.**
(TIFF)

**S4 Fig. Summary of risk of bias and applicability of RDTs.**
(TIFF)

**S5 Fig. Estimated sensitivity and specificity of ELISA tests by subgroups.**
(TIFF)

**S6 Fig. Predictive region of ELISA tests by subgroups.**
(TIFF)

**S7 Fig. Estimated sensitivity and specificity of RDTs by subgroups.**
(TIFF)

**S8 Fig. Predictive region of RDTs by subgroups.**
(TIFF)

**S9 Fig.** Predictive region in the global ROC space and after excluding the most influential studies for ELISA (A) and RDTs (B).
(TIFF)

## Author Contributions

**Conceptualization:** Sandra Helena Suescún-Carrero.

**Data curation:** Sandra Helena Suescún-Carrero, Philippe Tadger, Carolina Sandoval Cuellar, Laura Ximena Ramírez López.

**Formal analysis:** Sandra Helena Suescún-Carrero, Lluis Armadans-Gil, Laura Ximena Ramírez López.

**Funding acquisition:** Sandra Helena Suescún-Carrero, Carolina Sandoval Cuellar, Laura Ximena Ramírez López.

**Investigation:** Sandra Helena Suescún-Carrero, Philippe Tadger, Carolina Sandoval Cuellar, Lluis Armadans-Gil, Laura Ximena Ramírez López.

**Methodology:** Sandra Helena Suescún-Carrero, Lluis Armadans-Gil.

**Project administration:** Sandra Helena Suescún-Carrero, Laura Ximena Ramírez López.

**Resources:** Sandra Helena Suescún-Carrero, Carolina Sandoval Cuellar, Laura Ximena Ramírez López.

**Software:** Philippe Tadger.

**Supervision:** Sandra Helena Suescún-Carrero.

**Visualization:** Sandra Helena Suescún-Carrero, Laura Ximena Ramírez López.

**Writing – original draft:** Sandra Helena Suescún-Carrero, Philippe Tadger, Carolina Sandoval Cuellar, Lluis Armadans-Gil, Laura Ximena Ramírez López.

**Writing – review & editing:** Sandra Helena Suescún-Carrero, Philippe Tadger, Carolina Sandoval Cuellar, Lluis Armadans-Gil, Laura Ximena Ramírez López.

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
