## [Decision Letter · Decision Letter 0]

29 Jul 2022

Dear Mrs Suescún-Carrero,

Thank you very much for submitting your manuscript "Rapid Diagnostic Tests and ELISA for diagnosing Chronic Chagas Disease: Systematic revision and meta-analysis" for consideration at PLOS Neglected Tropical Diseases. As with all papers reviewed by the journal, your manuscript was reviewed by members of the editorial board and by several independent reviewers. The reviewers appreciated the attention to an important topic. Based on the reviews, we are likely to accept this manuscript for publication, providing that you modify the manuscript according to the review recommendations. 

We appreciate your presentation of a detailed systematic review and meta-analysis of studies assessing the test performance of ELISAs and RDTs for the diagnosis of chronic Chagas Disease. The methodology is well designed and the results have some clinical applicability. However, presentation of the results could be improved to enhance the flow and structure of the authors' findings.

Sincerely,

Eva Clark, M.D., Ph.D.

Section Editor

Nilson Zanchin

Section Editor

The authors present a detailed systematic review and meta-analysis of studies assessing the test performance of ELISAs and RDTs for the diagnosis of chronic Chagas Disease. The methodology is well designed and the results have some clinical applicability. However, presentation of the results could be improved to enhance the flow and structure of the authors' findings.

Reviewer's Responses to Questions

**Key Review Criteria Required for Acceptance?**

**Methods**

-Are the objectives of the study clearly articulated with a clear testable hypothesis stated?

-Is the study design appropriate to address the stated objectives?

-Is the population clearly described and appropriate for the hypothesis being tested?

-Is the sample size sufficient to ensure adequate power to address the hypothesis being tested?

-Were correct statistical analysis used to support conclusions?

-Are there concerns about ethical or regulatory requirements being met?

Reviewer #1: The objectives of the study are clearly articulated with a clear testable hypothesis stated. The study design is appropriate to address the stated objectives.

The Abstract mentions that the search was performed for studies published between May and August 2020, but the search was performed between May and August 2020, and included studies published between 2010 and 2020.

In Eligibility Criteria, the authors said that “The search included studies that estimated sensitivity, specificity, and predictive values…”. However, the predictive values have not been shown or analyzed in the Results.

Data extraction definitions. The presentation of the different groups to classify subjects is confusing and individuals may be in different groups. I believe that this classification could change to endemic/non-endemic region without losing relevant information.

Especially for RDT, it might be relevant to analyze whether the test was performed by skilled or unskilled health workers.

Reviewer #2: (No Response)

Reviewer #3: -Are the objectives of the study clearly articulated with a clear testable hypothesis stated? YES

-Is the study design appropriate to address the stated objectives? NEUTRAL. Please refer to commentary below. 

-Is the population clearly described and appropriate for the hypothesis being tested? YES

-Is the sample size sufficient to ensure adequate power to address the hypothesis being tested? N/A

-Were correct statistical analysis used to support conclusions? YES

-Are there concerns about ethical or regulatory requirements being met? N/A

The objective of the study is clearly indicated in the manuscript, although in the abstract please include the time lapse of publication (10 years) that was considered to include the studies (as this information if more important for the potential reader) instead of the time (May to August 2020) that the search was carried out.

**Results**

-Does the analysis presented match the analysis plan?

-Are the results clearly and completely presented?

-Are the figures (Tables, Images) of sufficient quality for clarity?

Reviewer #1: The analysis presented match the analysis plan, the results are clearly and completely presented. The figurs are of sufficient quality for clarity.

Reviewer #2: (No Response)

Reviewer #3: -Does the analysis presented match the analysis plan? YES

-Are the results clearly and completely presented? YES. Although it is advisable to review English as some phrases have a construction that can be confusing for the reader. 

-Are the figures (Tables, Images) of sufficient quality for clarity? YES

**Conclusions**

-Are the conclusions supported by the data presented?

-Are the limitations of analysis clearly described?

-Do the authors discuss how these data can be helpful to advance our understanding of the topic under study?

-Is public health relevance addressed?

Reviewer #1: The conclusions are supported by the data presented; the limitations of analysis are clearly described; the authors discuss how these data can be helpful to advance our understanding of the topic under study.

Reviewer #2: (No Response)

Reviewer #3: -Are the conclusions supported by the data presented? YES

-Are the limitations of analysis clearly described? YES

-Do the authors discuss how these data can be helpful to advance our understanding of the topic under study? YES

-Is public health relevance addressed? YES

**Editorial and Data Presentation Modifications?**

Reviewer #1: Reference 20 does not follow the required format for Plos NTD.

Reviewer #2: (No Response)

Reviewer #3: (No Response)

**Summary and General Comments**

Reviewer #1: Regarding the study entitled “Rapid Diagnostic Tests and ELISA for diagnosing Chronic Chagas Disease: Systematic revision and meta-analysis”, I think the authors have done a good work. The diagnosis of Chagas disease, particularly in the chronic phase, is based on the detection of specific IgG antibodies against Trypanosoma cruzi, with the ELISA test being the most common in laboratories, and immunochromatographic tests or rapid diagnostic tests (RDT) the most common in primary health care centers or field screening studies. In this framework, the authors summarize the available evidence on the diagnostic validity of ELISA and RDT in individuals with suspected chronic Chagas disease. The manuscript is easy to read and the results are straightforward. The conclusions of the work are relevant for public health. However, minor corrections are needed to make the article fit for publication in the Plos NTD.

Reviewer #2: Authors perform a systematic review and meta-analysis of studies assessing the sensitivity and specificity of enzyme-linked immunosorbent assay (ELISA) and Rapid Diagnostic Tests (RDT) among individuals with suspected chronic Chagas Disease. The methodology is strong and well designed with interesting results that would have some clinical applicability. The display of the results can be organized better to enhance the flow and structure of the information:

* Paper has too many figures. Please choose 3-4 to include in the main paper and move the rest to the supplemental data, especially the quality assessment data

* Key tables were moved to the supplement. Paper needs a table 1 with all the important information from all studies. Tables s4, s5, and s6 need to appear in the main manuscript as a single table or 2 tables. Please incorporate as much info as possible

* Even though it was stated in the method section, no information was provided for the heterogeneity and I2 scores data in the text. Please include and discuss

* Subgroup analysis did not provide p-values for each analysis. Where the different subgroups statistically different? Was the heterogeneity better with each of those subgroups?

* Line 342: sentence: "There were studies with atypical values (10,40,54) (Figure 8)" what atypical means? Please clarify

Reviewer #3: (No Response)

PLOS authors have the option to publish the peer review history of their article (what does this mean?). If published, this will include your full peer review and any attached files.

Reviewer #1: No

Reviewer #2: Yes: Andrés F. Henao-Martínez

Reviewer #3: No

Figure Files:

Data Requirements:

Reproducibility:

References

---

## [Decision Letter · Decision Letter 1]

3 Oct 2022

Dear Mrs Suescún-Carrero,

We are pleased to inform you that your manuscript 'Rapid Diagnostic Tests and ELISA for diagnosing Chronic Chagas Disease: Systematic revision and meta-analysis' has been provisionally accepted for publication in PLOS Neglected Tropical Diseases.

Best regards,

Eva Clark, M.D., Ph.D.

Section Editor

Nilson Zanchin

Section Editor

Reviewer's Responses to Questions

**Key Review Criteria Required for Acceptance?**

**Methods**

-Are the objectives of the study clearly articulated with a clear testable hypothesis stated?

-Is the study design appropriate to address the stated objectives?

-Is the population clearly described and appropriate for the hypothesis being tested?

-Is the sample size sufficient to ensure adequate power to address the hypothesis being tested?

-Were correct statistical analysis used to support conclusions?

-Are there concerns about ethical or regulatory requirements being met?

Reviewer #1: -Are the objectives of the study clearly articulated with a clear testable hypothesis stated? YES

-Is the study design appropriate to address the stated objectives? YES

-Is the population clearly described and appropriate for the hypothesis being tested? YES

-Is the sample size sufficient to ensure adequate power to address the hypothesis being tested? YES

-Were correct statistical analysis used to support conclusions? YES

-Are there concerns about ethical or regulatory requirements being met? YES

Reviewer #2: (No Response)

**Results**

-Does the analysis presented match the analysis plan?

-Are the results clearly and completely presented?

-Are the figures (Tables, Images) of sufficient quality for clarity?

Reviewer #1: -Does the analysis presented match the analysis plan? YES

-Are the results clearly and completely presented? YES

-Are the figures (Tables, Images) of sufficient quality for clarity? YES

Reviewer #2: (No Response)

**Conclusions**

-Are the conclusions supported by the data presented?

-Are the limitations of analysis clearly described?

-Do the authors discuss how these data can be helpful to advance our understanding of the topic under study?

-Is public health relevance addressed?

Reviewer #1: -Are the conclusions supported by the data presented? YES

-Are the limitations of analysis clearly described? YES

-Do the authors discuss how these data can be helpful to advance our understanding of the topic under study? YES

-Is public health relevance addressed? YES

Reviewer #2: (No Response)

**Editorial and Data Presentation Modifications?**

Reviewer #1: Replace in all the text "PDR" by "RDT".

In the footnote of Table 1, it is written "...table 2x2 shown in in this article".

In References, the correct way to abreviate the journal names is: 34. Trop Med Int Health; 72. Medicina (B Aires)

Reviewer #2: (No Response)

**Summary and General Comments**

Reviewer #1: The manuscript has been properly improved and in this new round I have no corrections or comments to make on it.

Reviewer #2: (No Response)

PLOS authors have the option to publish the peer review history of their article (what does this mean?). If published, this will include your full peer review and any attached files.

Reviewer #1: No

Reviewer #2: **Yes: **Andres Henao-Martinez

---

## [Editor Report · Acceptance letter]

13 Oct 2022

Dear Mrs Suescún-Carrero,

We are delighted to inform you that your manuscript, "Rapid Diagnostic Tests and ELISA for diagnosing Chronic Chagas Disease: Systematic revision and meta-analysis," has been formally accepted for publication in PLOS Neglected Tropical Diseases.

Best regards,

Shaden Kamhawi

co-Editor-in-Chief

Paul Brindley

co-Editor-in-Chief
